# Irruption and Absorption: A ‘Black-Box’ Framework for How Mind and Matter Make a Difference to Each Other

**DOI:** 10.3390/e26040288

**Published:** 2024-03-27

**Authors:** Tom Froese

**Affiliations:** Embodied Cognitive Science Unit, Okinawa Institute of Science and Technology Graduate University (OIST), 1919-1 Tancha, Onna-son, Okinawa 904-0495, Japan; tom.froese@oist.jp

**Keywords:** agency, consciousness, observer, mind–body problem, mental causation, neural entropy, compression, information creation, information loss

## Abstract

Cognitive science is confronted by several fundamental anomalies deriving from the mind–body problem. Most prominent is the problem of mental causation and the hard problem of consciousness, which can be generalized into the hard problem of agential efficacy and the hard problem of mental content. Here, it is proposed to accept these explanatory gaps at face value and to take them as positive indications of a complex relation: mind and matter are one, but they are not the same. They are related in an efficacious yet non-reducible, non-observable, and even non-intelligible manner. Natural science is well equipped to handle the effects of non-observables, and so the mind is treated as equivalent to a hidden ‘black box’ coupled to the body. Two concepts are introduced given that there are two directions of coupling influence: (1) *irruption* denotes the unobservable mind hiddenly making a difference to observable matter, and (2) *absorption* denotes observable matter hiddenly making a difference to the unobservable mind. The concepts of irruption and absorption are methodologically compatible with existing information-theoretic approaches to neuroscience, such as measuring cognitive activity and subjective qualia in terms of entropy and compression, respectively. By offering novel responses to otherwise intractable theoretical problems from first principles, and by doing so in a way that is closely connected with empirical advances, irruption theory is poised to set the agenda for the future of the mind sciences.

## 1. Introduction

Consider the event of you reading this text—is it part of reality? I would argue, yes, and in the full complexity of that event. It is reasonable to accept that your whole action is real—certainly your body posture, eye movements, etc., but equally certainly also your intention to read, your changing understanding, etc. This complex reality of our being in the world is described by two highly successful theories, physics and phenomenology, that focus on its material and experiential ontology, respectively. For over a century, these two theories have been at odds with each other, each claiming its own domain to be the more fundamental one. However, as we saw briefly with the example of reading, we do not encounter such a stark ontological opposition in our lifeworld; we normally engage in reading by being both an object in the world and a subject for whom that world shows up. Hence, a more sensible strategy might be to relax our definition of naturalism [1] and to accept and accommodate the existence of both ontologies [2]: mental phenomena are real and so are material phenomena, even if they are not of the same kind. 

If such an integration under a more relaxed naturalism with a sufficiently broad notion of reality is on the right track, we would expect that recognizing and respecting each of their ontological specificities will lead to a more productive scientific perspective, too. Ideally, it should be able to resolve the mind–body problem, at least in the form in which it has confronted modern science since its Cartesian origins, and perhaps even going back to the times of the ancient Greeks [3]. Indeed, a principal motivation for the founding of cognitive science in the 1970s was the ambition to create an interdisciplinary context in which to formally solve this age-old problem [4], which finally seemed to be within reach following advances during the preceding cybernetics era (e.g., [5]). 

Yet today, even after over half a century of concerted efforts, the mind–body problem has only become more entrenched, as its full extent has become articulated into at least two interdependent fundamental problems: the problem of mental causation and the problem of consciousness [6,7], as illustrated in Figure 1. To be fair, there exists a broad variety of proposed solutions, but these are widely recognized to remain incomplete, especially due to the “hard” problem of consciousness [8]. An intuitive way of capturing the key puzzle is to ask: *How could a material body make a difference to our subjective experience, and how could our subjective experience as such make a difference to the material body?*

This set of problems is no longer limited to the emergence of consciousness and its physical efficacy. Arguably, the same problems can be posed in a much more general form, as illustrated in Figure 2: how could a material body originate mental content [9], and how could that mental content *as such* make a difference to that material body [10]?

Let us briefly consider an illustrative example: in cognitive neuroscience, it has long been standard practice to appeal to neural activity as the vehicle of mental representations, such as place or grid cells, yet there is no consensus on how neural activity achieves the status of carrying intrinsic mental content. Moreover, even if we were to accept an account of the origins of this mental content for the sake of argument, the next problem would be that there is no account of how this content *as such*, that is, specifically as mental content, could make a difference to the dynamics of neural activity [11]. In fact, it is not even clear what we should be measuring: “Beliefs, desires, and so on are, after all, invisible. We see (what we take to be) their effects. But no one has ever actually seen a belief. Such things are (currently? permanently?) unobservable” (Clark [12], p. 3).

We will not go further into this lively debate about problems related to mental content, but a healthy dose of skepticism is justified: “As things stand, it is far from clear that it will prove possible to solve these problems” (Hutto and Myin [13], p. 83). In other words, if we can speak of progress at all, it is rather in the sense that is becoming clearer just how intractable and prevalent the implications of the mind–body problem really are.

**Figure 1 entropy-26-00288-f001:**
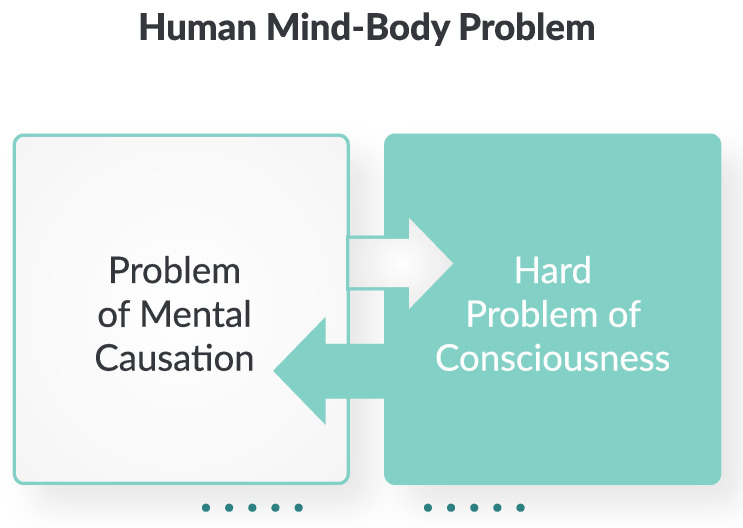
The mind–body problem refers to the problematic relationship between physical properties, as measured by the natural sciences, and mental properties, as investigated by the humanities. In accordance with the bidirectionality of this relationship, the mind–body problem can be recast as two intertwined sub-problems in the philosophy of the mind [6]: the problem of mental causation (mind to matter), and the hard problem of consciousness (matter to mind).

**Figure 2 entropy-26-00288-f002:**
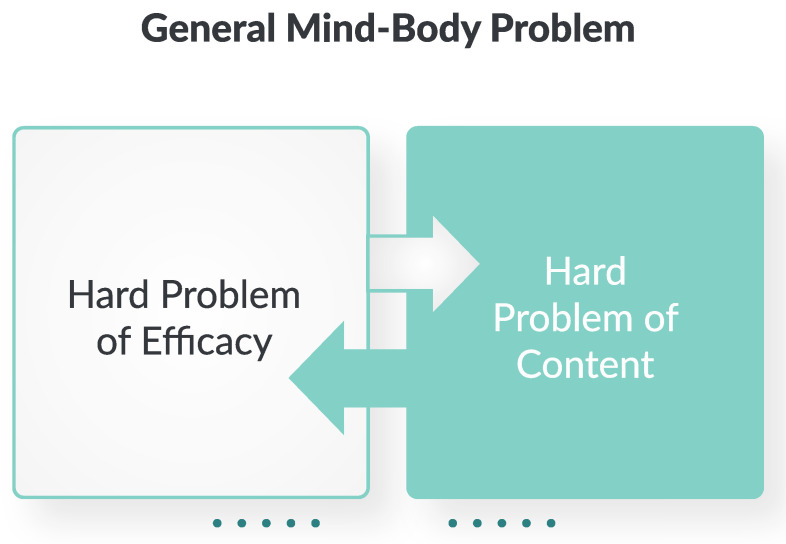
The classic mind–body problem can be generalized such that it also applies to other forms of life, which exhibit agency and subjectivity to varying degrees [14]. Hence, this generalized mind–body problem can be recast as two intertwined sub-problems: the hard problem of efficacy (mind to matter) [10] and the hard problem of content (matter to mind) [9].

At the same time, the overall scientific stakes are getting higher: this lack of a breakthrough with respect to the mind–body problem in its different manifestations is starting to stymie progress on fundamental problems in physics because its best theories are being limited by unclear conceptualizations of the role of the observer in measurement and the material universe more generally [15]. Consider how theoretical physicists implicitly adopt the classic internalist-representationalist response to the mind–body problem as a starting point of their own work: “A hard lesson to learn is that our sensations are partly caused by reality, but are fully constructed by our brains to present the world to us in just the form we need to make our way in nature” (Smolin [16], p. xiii). What would be the consequence for physics if this problematic account of the origins of perceptual content turns out to be misguided [17]? What if we need to rethink the relation between the material and mental by e.g., acknowledging that intentions and experiences cannot be “fully” reduced to brains in the way that Smolin envisions here?

So far, cognitive science sadly does not have much to contribute to this debate on the role of mind at this fundamental level, which is odd considering how much importance it otherwise assigns to whether any putative theory of mind is *consistent* with physics–even though which kind of physics is used as the criterion is often left rather vague. The deeper issue may be that, implicitly, this consistency criterion is often interpreted in a much more conservative sense that a theory of mind cannot impose any additional conditions on a theory of matter. However, this has the undesirable consequence of making cognitive science rather powerless to have a genuinely two-way dialogue with physics. Ideally, a more productive theory of mind should also serve to help contemporary physics to better arbitrate between its many otherwise largely equivalent theories on the nature of reality.

In summary, given the high stakes and the stalled progress, there is an urgent need for fresh theoretical perspectives on the mind–body problem. The pressure is mounting to overcome deeply entrenched assumptions and ideally to do so in a testable manner. In the following section, we will leverage the recently proposed “irruption theory” [18] to take initial steps toward opening up this new scientific horizon.

## 2. Irruption Theory as a “Black-Box” Framework

Everyone can easily verify the bidirectional interaction between mind and matter for themselves: we act for reasons and because things matter to us, and in turn the material world shapes how we feel and what we perceive. Consider a trivial example: I perceive a tension in my upper back; therefore, I stretch my arms, and with this change in the posture of my material body, I start to relax and feel better. Let us begin by accepting this kind of example at face value: both mind and matter are real, and they make a real difference to each other. However, scientifically, the key issue is that the mind–matter relation itself is not given to our observation; hence, it is not evident precisely *how* this difference-making happens. The dominant scientific strategy to get around this issue has been to search for ways in which the subjective mind can be conceptualized as just another aspect—function, organization, information, etc.—of observable matter. Arguably, however, that kind of conversion is not possible without simultaneously losing precisely that which is specific to the domain of the subjective mind.

These conceptual limitations are most evident in the case of consciousness: “What is ‘left out’ by adopting an exclusively physical view of things is the possibility of describing the character of experience (the phenomenology problem) and the possibility of having an *intelligible* understanding of how experiences can be ‘understood’ in such terms (the metaphysical problem)” ([19], p. 48). Accordingly, Hutto argues that the hard problem of consciousness should be cast as a problem about *intelligibility*, and accepts that in this form it cannot be solved, not even in principle [19].

At first sight, accepting unintelligibility may appear to be an unattractive proposition because it seems to imply that we might as well give up on making progress. On the other hand, getting clearer on the limits of what is scientifically actually possible in principle may be precisely what is needed to finally start making genuine progress in the direction where progress can be made. For instance, if we further convert the conceptual problem of intelligibility into methodological problems of predictability, we can take steps toward developing alternative testable hypotheses about the material signatures of bidirectional mind–body interaction, as illustrated in Figure 3.

Note that, following from our starting point of realism about both mind and matter, this points to an intrinsic unpredictability that does not depend on our measurement of the material events that are associated with mental activity. In this view, mental activity does not leave the material world unchanged, yet we cannot rely on any assumption of mind–matter isomorphism, like a neural “bridge locus” [20]. It would even go too far to assume that the mind is identical to some formal system property, such as an order parameter attractor state or top-down constraint [21,22,23]. Rather, the mind’s most immediate impact on matter, that is, before it undergoes further transformations in the body, must be conceptualized as equivalent to a hidden ‘black box’. This has the notable advantage that we can employ the hard problems associated with the mind–body relation as a means of specifying from first principles the way in which the material world is affected by its coupling with the mind in terms of this ‘black-box’ middle, as illustrated in Figure 4.

Irruption theory expects that the exertion of agency and the arising awareness are necessarily accompanied by measurable changes in the material processes that are associated with it. Nevertheless, it rejects the traditional commitment to the *intelligibility* of such measurable changes. This is a radical change in perspective, yet also a necessary one. It is *radical* because it implies a relaxation of a core principle of the scientific method, namely the principle of understandability. It is also *necessary* because otherwise, that principle would join up with an even more fundamental scientific principle, the principle of objectification, to entail the exclusion of the conscious subject from the scientific worldview [3,24]. Such an exclusion of ourselves, as actors and observers, from the world image we are creating would be in tension with another scientific requirement: “theories must not deny the validity of observations. A theory can be scrupulously logical and predictive, but if it covers its own tracks, then it fails the standards of science” (Musser [15], p. 145). Hence, if we cannot reject the reality of the material world as captured by the principle of objectification and we cannot reject the reality of our subjective experience, then we are forced to accept the relaxation of the principle of understandability. This amounts to nothing but accepting the prospect that nature can appear strange to us, especially when we indirectly observe the material consequences of the subjective mind [10].

Importantly, this change in perspective toward a ‘black-box’ framework should not be mistaken for the kind of resignation from a scientific inquiry that goes under the label of “mysterianism”, and which many rightly consider a dispiriting conclusion that is to be strongly resisted ([15], p. 240). Rather, by accepting that there are fundamental limits on the very intelligibility of material changes that are due to agency and consciousness, we can launch a scientific research program aimed at formalizing and quantifying their material efficacy as a specific kind of uncertainty-inducing process of underdetermination.

Let us consider in more detail the two principal ways in which the matter of the body is impacted by the existence of the embodied mind. What are the key signatures that we should look for to evaluate whether a body is minded? For ease of exposition, the focus will be on the classic problems of mental causation and consciousness, but the essential points could be readily generalized to the hard problems of agential efficacy and mental content in future work.

### 2.1. Mental Causation

Let us return to the example of the stretch. I stretched my arms because I desired to relieve tension in my back, I had the belief that this action would make me feel better, and hence I changed my posture accordingly. But, as we noted, beliefs and other mental content are not directly observable in the material domain. So, what do we observe in place of that content? If we stay true to our realist premise about the mind, and hence accept that my desire *as such* made a difference *in its own right* to my action, then we are also forced to accept that this mental involvement manifested as changes in my body that cannot be completely derived from all preceding observable material causes, not even in principle.

This claim of non-derivability is somewhat akin to the famous “strong free will theorem” by Conway and Kochen [25], except that the presence of agential efficacy is not secondarily derived from a property of quantum physics, but rather directly from a priori considerations of the very nature of the mind–body relation. Specifically, we are led to hypothesize that, given that the arm stretch was not just a mere physical event, but also a motivated activity, at some scale of its material basis there must have correspondingly been an unintelligible change in hidden factors—the motivations that cannot be directly objectified—making a difference to its material unfolding. It is this background difference in increased unintelligible physical events, which mark the change as motivated, that we will refer to as *irruption*.

The concept of irruption into the material basis of motivated activity is amenable to formalization and quantification; as a first pass, it can be approximated by applying information-theoretic measures of entropy production to neurophysiological variables [18]. Irruption can be brought into connection with the dissipation of thermodynamic free energy due to its approximation by disordering tendencies such as noise [10]. This concept has the advantage that it avoids the traditional problem of accounting for a direct causal link from mind to matter, for example from intention to action. Instead, to account for the normally close fit of bodily action to mental intention, it appeals to a process of *attunement* [18]: the history of agent–environment interaction, in combination with a suitable notion of plasticity, can serve as the basis for explaining the otherwise surprising appropriateness of our actions [26]. In other words, the role of irruptions is simply to indirectly widen the space of action possibilities, while the process of action selection is offloaded into the self-organizing propensities of our body and affordances of the environment [27].

In addition, the process of attaining attunement itself could be facilitated by irruptions. Connectionist models of minimal complex adaptive systems have shown that the alternation of divergent and convergent state tendencies, in combination with a basic form of associative memory, can give rise to global constraint satisfaction and generalization without centralized control or supervision (e.g., [28]). This form of errorless learning was first introduced into the literature as optimization by means of “self-modeling” [29], but the term “self-modeling” has also been popularly adopted by error-based predictive processing accounts of agency and consciousness (e.g., [30]). Hence, to highlight this crucial distinction with this class of implicitly or even explicitly representational accounts, irruption theory has adopted the terminology of the “self-optimization” model [31].

More generally, the application of entropy measures in cognitive science is already an active field of research, and this is not the place to conduct a systematic review of the evidence. Importantly, strong conceptual links between the expansion of organisms’ cognitive capacities over evolutionary and developmental time and an increase in entropy production have already been proposed [32,33]. Additionally, methodological proxies for entropy production, such as broken detailed balance, irreversibility, and state diversity, are being developed to track levels of cognitive effort and conscious awareness [34,35,36].

The current proposal of viewing neural entropy through the lens of irruption theory offers a more nuanced interpretation of this accumulating evidence. For example, in the context of stochastic accumulator models of voluntary movement, an open conceptual challenge is that “theorists who want to identify the source of action as the agent will have to tell a story that somehow makes a case for the noisy trigger being part of or attributable to the agent” ([37], p. 566). The concept of irruption fits this requirement well. Starting from first principles, irruption is defined as any physiologically unintelligible change in bodily activity associated with a manifestation of agential efficacy, which hence will appear akin to a change in noise correlated with the exertion of volition.

In this way, irruption theory could be developed into a novel framework for addressing the classic problem of free will: on the one side, it aligns with theorists appealing to the role of indeterminacy in agency for opening up a space of action possibilities (e.g., [38,39]), while on the other side, it also accepts, along with determinists (e.g., [40]), that we are not in direct control over which action is finally selected. Under normal conditions, we are free to increase (or not) our space of action possibilities, but we cannot directly control the behavioral outcome—we trust in the self-organizing processes out of which an action will actualize, which involves spontaneous coordination of internal possibilities in accordance with environmental constraints and affordances. An adequate history of self-optimization ensures that most of the time such agent–environment interaction works out as expected.

Finally, irruption theory enables us to reduce some of the interpretational ambiguity associated with measures of neural state divergence. For instance, they have occasionally been taken as indicative of a higher level of conscious awareness (e.g., [41]), but this may turn out to be confounding that higher level of awareness with the increased agential efficacy that is also enabled by that level of awareness. As we will discuss in detail below, for irruption theory, conscious experience as such is associated with its own distinctive material signature, which is defined by the concept of *absorption*.

### 2.2. Conscious Experience

We can now consider how irruption theory enables us to reconceive the other direction of mind–matter interaction, specifically the hard problem of consciousness [8]: How can our material body make a difference to our conscious experience? From the traditional perspective of cognitive science, the relation between consciousness and its material basis is characterized by supervenience, which is a one-way dependence of consciousness on its material basis [6]. On this traditional view, it does not make any difference to a material process whether it is making any difference to the phenomenology of a conscious experience; when we are observing its material basis, the conscious experience that supervenes on it might as well be epiphenomenal or even inexistent. Recent work has started to take a different stance, arguing that for an agent’s consciousness to be detectable by us in principle, it must have some observable effect on the material universe, for example in terms of covert action or other active states that intervene causally [30,42,43]. Simply put, the presence of conscious experience cannot leave the material world unchanged.

Irruption theory broadly aligns with this interactionist stance, yet as argued above, it takes a more radical stance by rejecting the commitment to an intelligible bridge locus. It thereby partially relaxes the scientific principle of understandability: all we can say is that when a material process makes a difference to a conscious experience, this is effectively akin to that material process making a difference in a hidden ‘black box’. Accordingly, there will be a corresponding unintelligible decrease in that material process’ observable efficacy to create material differences. This is somewhat akin to the proposal of relational constraints as the material basis for mental content, in that both of them also cannot be directly measured [44]. However, irruption theory goes further by specifying that there is an irreducible element of unintelligibility: the material process that is making a difference to the subjective mind undergoes an unpredictable reduction in their otherwise expected efficacy to make material differences, for which we introduce a novel term, *absorption*.

The concept of absorption can be approximated by applying information-theoretic measures of redundancy, such as compression. Again, the use of such measures is already an active field of research in cognitive neuroscience [45] and this is not the place to conduct a review. Importantly, there is a broad consensus that a key signature of consciousness in the brain is that its high-dimensional neural state space is transformed into a low-dimensional system [46]. In terms of information theory, this phenomenon has been expressed in terms of the following thesis: “The feeling of qualia is the result of an efficient compression of information about prior experiences”. (Jost [47], p. 8). Relatedly, it has been argued that the material basis of consciousness is characterized by information closure [48], which can be interpreted as a measure of decreased specific variability due to absorption. Similarly, a popular dynamical systems approach is to capture the neural dynamics associated with subjective experience in terms of the emergence of low-dimensional collective order parameters [22,49].

It is worth highlighting again that the concept of absorption does not offer a solution to the hard problem of consciousness in the traditional sense of an explanation of the origins of subjective experience that is intelligible in terms of material processes. Instead, it turns the current explanatory shortcomings into an intrinsic aspect of the phenomenon that needs to be explained. Consider the following skeptical assessment of the state of the art of the field of consciousness science: “None of these theories tell us why or how the binding of information—through association, oscillation, synchronisation, re-entry, massive integration and the like—should *necessarily* give rise to experience”. (Solms [7], p. 85). This is intended to sound like a wide-ranging indictment of the theories and yet, from the perspective of irruption theory, the situation could not be otherwise. We should not expect an intelligible entailment from matter to consciousness in the first place.

Fortunately, we do not need to view this relative relaxation of the principle of understandability as the end of scientific inquiry; rather, the hidden ‘black-box’ framework enables us to derive testable hypotheses about the material basis of consciousness from first principles, namely the concept of absorption. Paradoxically, it is by accepting strict limits on intelligibility that we can come to properly understand why the material basis of consciousness tends to take the specific forms it tends to take. Absorption can be measured in terms of the binding of information, which thereby becomes the measurable correlate of a material process impacting the mind via a ‘black-box’ middle.

## 3. Discussion

Irruption theory interlinks two traditionally separate domains of investigation into a unified framework that still manages to respect their distinctive characteristics. It thereby provides fertile ground for further developments. A few directions for future research are outlined below.

### 3.1. Spatiotemporal Complexity

A key implication of irruption and absorption is that there is an irresolvable tension between the material basis of mental causation and consciousness and in general between all forms of agential efficacy and mental content. Irruption theory therefore aligns with existing requirements that any material system embodying a conscious agent must be a composite system [42], although it makes unique predictions. For instance, while it accepts that action and perception are closely integrated into the bidirectional relation between mind and matter, it cautions against identifying action and perception too closely (e.g., [50]). Action and perception are expected to have contrary effects on the body: whereas action involves the irruption of differences *into* a material process, perception involves the absorption of differences *from* a material process. These competing effects of sensorimotor interaction should have measurable consequences, for example in terms of changes in the overall level of specific variability over time. We can hypothesize that irruption coincides with decreases in shared variance of physiological processes, whereas absorption coincides with increases in their shared variance.

Accordingly, the competing requirements for action and perception are expected to interfere with each other, given that there cannot simultaneously be both a decrease and an increase in shared variance, at least not in the same place. Irruption theory, therefore, predicts two essential design principles of sensorimotor systems: (1) material processes involved in action and perception optimally do not directly overlap in *space*, which gives a new perspective on the anatomical separations of the brain—motor regions are indeed mostly distinct from sensory regions; (2) material processes involved in action and perception optimally do not directly overlap in *time*, especially if they are otherwise overlapping in space, which gives a new perspective on the prevalent rhythmicity of many neural and other organismic processes across temporal scales.

In this way, irruption theory provides the conceptual spatiotemporal foundation from which to develop a host of fresh perspectives on the organization of the organism and the brain. For example, given the competing material effects of irruption and absorption, theoretical work on grounding a minimal temporal granularity in the organismic basis of agential efficacy [51] and in the large-scale neural synchrony leading to conscious experience [52] could become integrated into a unified account of temporality. A first prediction derivable from this unified account is that lived temporality cannot be independent of motivated activity, which suggests there could be a fruitful encounter between irruption theory and existing literature on intentional and temporal binding [53].

### 3.2. Biological Organization

Irruption theory predicts that organisms exhibit distinct categories of physiological characteristics related to irruption-dominant and absorption-dominant processes, namely those that facilitate spontaneous diversification into a high-dimensional regime and those that facilitate collapse into low-dimensional redundancy.

In this regard, irruption theory aligns well with neuroscientific research into the role of traveling waves for the brain’s functional systems (e.g., [54]). A traveling wave could be interpreted as providing an optimal solution to the incompatible spatiotemporal markers of irruption and absorption: it has been proposed that the function of this phenomenon has to do with spontaneous switching between moments of segregation and large-scale integration. More specifically, it has been hypothesized that conscious experience occurs in aperiodic cycles, with the moments of conscious experience coinciding with integration events [55], just as would be predicted by irruption theory.

More generally, the opposite requirements of irruption and absorption nicely line up with existing proposals in theoretical biology, which highlight that organisms stand out from non-living systems both for their *divergent* tendencies, as captured by such concepts as extended criticality, symmetry breaking, and non-ergodicity, and for their *convergent* tendencies, as highlighted by such concepts as anti-entropy, symmetry, and relative invariance [56]. These tendencies can now be given novel theory-driven interpretations in terms of the competing tendencies of irruption and absorption, which would be entailed by the efficacy of the embodied mind from the origins of life onwards.

Accordingly, it is likely that this ‘black box’ framework of the mind–matter relation could be adapted to account for the top-down and bottom-up cross-scale couplings that integrate the organism into a unified phenomenon within the domain of matter. In the context of basal cognition research, these two directions of cross-scalar influences between cells and their collectives are already suitably described in terms of the *deformation* of normal activity patterns and the *partial erasure* of individual identity, respectively [57]. This makes it plausible that irruption theory could be generalized to account for interactions across ontologies nested within the domain of matter. The best conceptual fit for irruption theory is to be expected for cross-scalar influences in the organism that coincide with ontological boundaries between distinct individuals at different scales [58], which would suggest that these individuals are related in an efficacious yet mutually unintelligible relation akin to the mind–matter relation.

### 3.3. Phenomenology of the Embodied Mind

The focus has been on the side of the natural sciences, but irruption theory also connects well with core insights in the phenomenological tradition. For instance, it is well known that being engrossed in a flow of activity goes hand in hand with a reduction in self-consciousness, a phenomenon that has been analyzed extensively in terms of “*absorbed* coping”, and which is, to some extent, even describable as a mindless activity [59]. Recalling the two paths mediated by the ‘black-box’ middle in Figure 4, irruption theory also expects that such a commitment to embodied activity has several notable consequences: (1) an increase in the fluidity and diversity of body movements (due to elevated irruption into matter); (2) a decrease in the degrees of freedom of mental activity (due to elevated absorption of mind); and, as a side-effect, (3) a decrease in subjective awareness (due to elevated irruption into matter interfering with its absorption).

Now consider what would happen if a skilled expert in a flow state were to engage in explicit monitoring of action execution: this would entail a shift from the lefthand side to the righthand side of Figure 4, coinciding with (1) a decrease in fluidity and diversity of body movements (due to decreased irruption into matter); (2) an increase in subjective awareness (due to elevated absorption from matter to mind); and, as a side-effect, (3) a decrease in absorbed coping (due to elevated irruption into the mind, causing interference with its absorption). In small doses, these consequences are akin to a perturbation-induced shift from readiness- to unreadiness-to-hand [60]. However, when self-monitoring is done too much, these consequences cause a breakdown in skilled performance, which is a phenomenon well-known in sports psychology as “choking” [61].

This raises the question of how we should conceive of the phenomenon of absorption from matter that coincides with an irruption into the mind. As a starting point, it is useful to note that there is a consensus in the phenomenological tradition that material processes of our embodiment are constitutive of a kind of passivity and opaqueness in our conscious experience [62]. The primary impact of embodiment is therefore taken to be experienced as a spontaneous and unintelligible increase in affectivity, which is somewhat intrusive yet at the same time can become a resource for mental change. The notion of “irruption” has been independently introduced to describe precisely this kind of influence:

“When bouts of unanticipated intensity well up within routine activity, they provide an occasion for change, potentially inspiring fresh articulations of what seemed self-evident before. Affect in this sense is a generative *irruption*, potentially kindling transitions from established understandings toward new thoughts and new discursive and practical moves. What is at issue is a dynamic reservoir of possibility, spheres of potential—what is formative but not yet formed”. (Slaby and Mühlhoff [63], p. 38; emphasis added).

Future work could unpack the phenomenological side of irruption theory in more detail. As a further motivation for undertaking that development, there is an opportunity to advance the original ambitions of the neuro-phenomenology project in the context of this novel “black-box” framework [64].

### 3.4. Additional Opportunities for Theory Development

Irruption theory holds promise to integrate across vast areas of knowledge. Here are some additional speculative topics that are worthy of more careful consideration:The concept of absorption of physiological activity is overly crude at the moment, as it encompasses everything from how the world impacts us effectively to how making contact with an object contributes to veridical perception. These absorption processes could be differentiated into a hierarchy of nested timescales [65], which unfold in a circular fashion [66].For the case of object perception, it may be possible to develop a notion of precision of absorption, similar to how precision has been employed by predictive processing accounts of conscious experience [30]. The standard account holds that perceptual content is specified by top-down predictive signals, with its veridicality possibly constituted by the counterfactual richness of their generative models [67]. In the current view, such signals may rather serve to absorb specific aspects of incoming sensory activity, thereby contributing to a detailed perceptual presence.There is nothing preventing us from tracing the incoming sensory activity out of the brain, into the body, and eventually into the environment: what is absorbed is this entire causal chain or world line. This appeal to the role of the environment evokes ecological psychology, which holds that it is the specificity of our relation to objects that constitutes our perceptual experience of their properties [68]. Accordingly, it may be possible to integrate suitably modified versions of predictive processing and ecological psychology into a theory of absorption: the medium outside the organism is already sufficiently rich in ambient information, which a suitably poised brain can absorb with high precision (see also [69]).The concept of absorption might also put us in a better position to make sense of otherwise puzzling features of nervous system organization. Consider what has been termed the “principle of undifferentiated encoding”, which emphasizes that receptor cells do not specify the physical nature of what caused their response [70]. Similarly, despite the vast complexity of activity arriving at a neuron, neural response is mainly governed by an “all-or-none law”. It seems wasteful for the material basis of perception to throw out all of this information at every step—unless the aim is precisely to absorb material variability as efficiently as possible.

## 4. Concluding Remarks

As the preceding discussion has illustrated, irruption theory provides fertile ground for a wide range of additional theoretical and methodological developments. In terms of the latter, an important topic for future research will be the mathematical formalization of the concepts of irruption and absorption, such that the most suitable methods for their measurement can be identified and further developed. Fortunately, there already exists an active community of researchers with overlapping methodological interests. Irruption theory shines brightest in terms of what it can potentially offer with respect to the development of new theoretical perspectives.

As it stands, cognitive science is beset with a number of fundamental anomalies deriving from the unsolved mind–body problem. Consciousness and free will are among the most widely known problems, but they are joined by a variety of related problems having to do with the very foundations of cognitive science, including mental causation, mental content, and agential efficacy. The essential move is to stop struggling against these long-standing theoretical problems in the vein hope of finally solving them for good, and instead to accept—and to start working with—what these struggles have already so strongly suggested: the mind–body relation is characterized by an intrinsic uncertainty. Once we make our peace with the consequent need to also relax the demands of the scientific principle of understandability at the scale of the organism, a new horizon of research opportunities opens up for cognitive science. By accepting strict limits on observability and intelligibility, the field can better benefit from a rich toolbox of existing methods that have been designed to work with the uncertainties that already abound in the natural world, such as by employing a ‘black-box’ framework. Cognitive scientists will have to overcome their distaste for unintelligibility; as the quantum revolution so famously demonstrated, sometimes measurement uncertainty is a feature, not a bug.

Indeed, by specifying the distinctive material conditions of an effective mind–body relation based on first principles, irruption theory will make it easier for cognitive science to enter a productive dialogue with the rest of the natural sciences. For example, it would be worthwhile to take a closer look at the measurement problem in quantum physics from the starting point that absorption is a necessary correlate of observation. Stronger contact between the mind sciences and physical sciences at this fundamental description of nature could provide a more solid foundation from which to build up our understanding of the mind–body relation across increasing scales of observation.

## Figures and Tables

**Figure 3 entropy-26-00288-f003:**
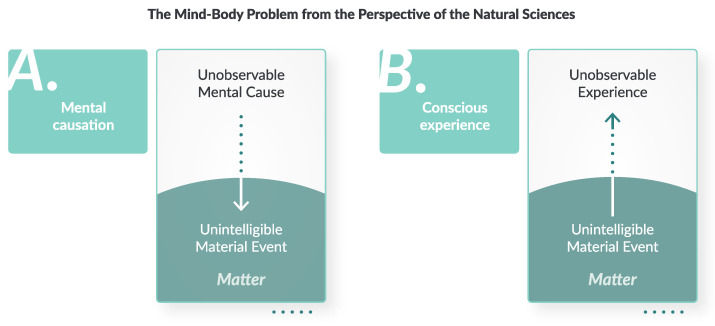
The mind–body problem as viewed from the perspective of the natural sciences. Mental content (**A**) and conscious experiences (**B**) are *unobservable* in the material world, and the consequences of their bidirectional relation with the material world are *unintelligible*. These in-principle limitations set an upper boundary on the predictability of material events that are related to mental activity; there is an intrinsic uncertainty associated with embodied action.

**Figure 4 entropy-26-00288-f004:**
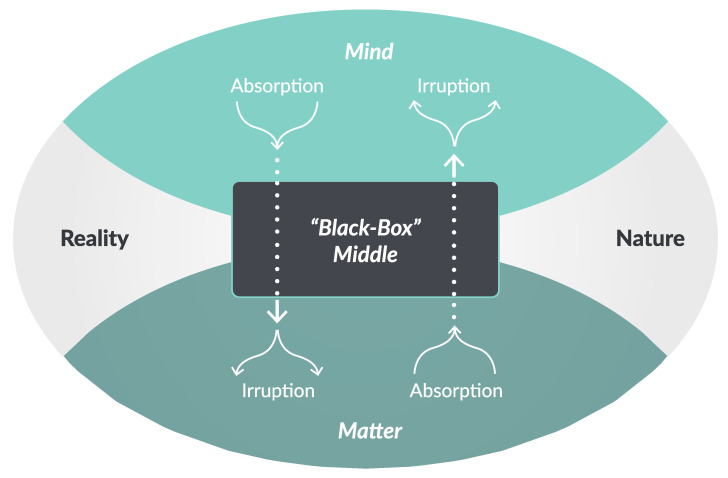
Ontology of irruption theory: mind and matter are part of one reality, but they are not the same. This complex relation of mind and matter enables the reality of their mutual interaction, while their irreducible ontological specificity entails mutual unintelligibility (equivalent to a “black-box” middle). Hence, when the mind makes a difference to the material world, this will manifest in the material world as an unintelligible increase in measurable differences (*irruption*). Similarly, when the material world makes a difference to the mind, this will again manifest in the material world, but in this case as an unintelligible decrease in measurable differences (*absorption*). Complementary considerations apply to how the mind–matter relation will manifest in the domain of the mind.

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
