# Peer review of "Irruption and Absorption: A ‘Black-Box’ Framework for How Mind and Matter Make a Difference to Each Other"

_entropy, 2024, doi:10.3390/e26040288_

Round 1
Reviewer 1 Report
Comments and Suggestions for Authors
Thanks for this paper, it is excellent and intriguing. No conceptual or content issues to contend with at all. Very satisfied that it meets publication standard in its current form. I only thought there was perhaps an omission in stating that the physical sciences made a giant leap forward when the uncertainty principle was accepted as a fundamental limit on observation/measurement (although it is mentioned in the final paragraph). To state that uncertainty or unintelligibility are complementary features, not bugs, seems like a strong contrast between the relative maturity of psychology (cognitive science and phenomenology) and physics. Just a suggestion.
Author Response
Thank you very much for this encouraging review!
It's indeed good to know that there is a precedent in the history of science where the formalization of uncertainty unleashed a profound revolution, namely quantum physics. However, for strategic reasons, the current proposal starts by motivating the formalization of unintelligibility from the foundational problems faced by cognitive science. It is prudent to avoid the impression that irruption theory is proposing another quantum approach to consciousness; in fact, it is offering a much more general theoretical framework. The fact that a revolution in cognitive science may also ultimately derive from a recognition and formalization of uncertainty inherent in the mind-body relation will make for fascinating future research.
Reviewer 2 Report
Comments and Suggestions for Authors
I will preface my comments with the admission that I did not fully appreciate this paper. There are many problems here, some of them are technical, while others are philosophical. I read the earlier paper in Entropy (the Introductory article on irruption theory), and while it was helpful, both articles do seem like slices from the same salami stick.
1) the approach to critique is over the top and ultimately not helpful.
* one alarm bell for me was the "radical" approach, which posits that if you simply ignore the tenets of naturalism, you can beat the system and reveal the truth of a (surprise!) non-material interpretation. It should not be unfair to say that this is a rather transparent end-run around the "science" part of CogSci.
* the critique of CogSci was not all that interesting. There are seemingly infinite critiques of reductionism, none of which directly improve upon reductionism. What is the historical and empirical justification of your approach, other than the pace of progress with respect to your preferred direction of Cognitive Science?
* it would be interesting to see how the discussion of the relevant 4Es (particularly enaction and embodiment) can cash out in the form of a toy or computational (agent) model.
2) the approach is not particularly rigorous: there are neither a toy model nor other type of computational model to convey how irruption might work.
* my interpretation of irruption is that it is an emergent process -- specifically that it emerges from interactions between neurons/regions of interest in the brain. Unlike the autonomic or so-called unconscious processes, irruption allows conscious processes to suddenly appear, which could be understood as an instance of strong emergence. I like the overall concept, but disagree about how we get there, or that it is not compatible with naturalism.
* there does not seem to be an information-theoretic formalization, which I would have expected from a submission to Entropy journal. This is actually disappointing, as there is a natural connection between characterizing emergent processes such as this and the information processing details of embodied cognition.
The balance of the paper is how this theory might fit into current debates in the popular literature, particularly the morass of consciousness and both sides (?!) of the free will debate. What might be more useful is how these fit into the Cognitive Science enterprise itself (such as it is). A few compelling questions: How is irruption theory useful for non-Philosophers? How can we operationalize irruption theory with respect to neuroimaging techniques? Is irruption theory a human-only phenomenon, or should be expected to see similar phenomena in animal cognition? What are the phylogenetic roots of irruption, and how does it relate to specific behaviors and cultural practices?
The toy/computational model aspect might be where the paper might benefit the most. Irruption theory could benefit from a formalization: computational agent models, connectionist/perceptron model, or something else that would demonstrate the conditions and circumstances under which irruption occurs. A dynamical systems model might also be good for describing the dynamics of an agent that has or has not experienced irruption.
The references section is actually pretty valuable. The references as cites somehow did not translate into a compelling presentation.
Author Response
I thank the reviewer for highlighting that in the introduction there was scope for potential misinterpretation of the critical target of irruption theory. I certainly did not intend to give the impression that the aim was to simply ignore the tenets of naturalism, or to do an “end-run” around the science part of cognitive science. To the contrary, the current proposal distinguishes itself from some “4E” cognition approaches by explicitly stating from the start that physicalism can be taken up as a valid, albeit incomplete, position. Phenomenology does not have the final say in this story, although it certainly does have a say too. Moreover, a key motivating idea is that specifying the extent of the incompleteness of the physicalist position with conceptual precision will give a significant boost to the science of cognitive science. I re-wrote the introduction to make these broadly naturalist commitments clearer.
To be fair, I did not offer an interesting review the history of arguments for and against reductionism, as this was not the place to do so. The main ambition was to sidestep those stalemates for a moment in order to show that it is possible to move the whole theoretical debate about non-reductionism into a more scientific arena. The rather practical idea was to propose that – to the extent that reductionism about embodied action falls short because physicalism is an intrinsically incomplete account of embodied action – this explanatory gap should itself be objectively detectable in terms of corresponding changes in the lower bound of the uncertainty of the measured physiological events.
The reviewer is right to complain about the lack of a computational model in the current presentation. This is a high priority for future work. Fortunately, I’m aware of at least three distinct groups of researchers developing different approaches, so hopefully there will be some relevant modeling literature available soon.
I don’t entirely agree with the reviewer’s complaint about the approach being not particularly rigorous. The conceptual tools introduced with this framework are specifically devised with their scientific applicability in mind. We are currently working on their mathematical formalization so hopefully there will be more in this direction coming out soon. The main ambition of this Hypothesis-type article was to present the full irruption theory framework, now complemented by the essential side of subjective experience and its associated concept of absorption, which can now be the target for model and method developers.